# Long-Term Effects of COVID-19 on Workers in Health and Social Services in Germany

**DOI:** 10.3390/ijerph19126983

**Published:** 2022-06-07

**Authors:** Claudia Peters, Madeleine Dulon, Claudia Westermann, Agnessa Kozak, Albert Nienhaus

**Affiliations:** 1Institute for Health Services Research in Dermatology and Nursing, University Medical Center Hamburg-Eppendorf, 20251 Hamburg, Germany; a.kozak@uke.de (A.K.); albert.nienhaus@bgw-online.de (A.N.); 2Department of Occupational Medicine, Hazardous Substances and Public Health, Institution for Statutory Accident Insurance and Prevention in the Healthcare and Welfare Services, 22089 Hamburg, Germany; madeleine.dulon@bgw-online.de (M.D.); claudia.westermann@bgw-online.de (C.W.)

**Keywords:** COVID-19, health workers, social workers, persistent symptoms, long COVID, post-COVID-19 syndrome

## Abstract

Health workers are at increased risk for SARS-CoV-2 infections. What follows the acute infection is rarely reported in the occupational context. This study examines the employees’ consequences of COVID-19 infection, the risk factors and the impact on quality of life over time. In this baseline survey, respondents were asked about their COVID-19 infection in 2020 and their current health situation. Out of 2053 participants, almost 73% experienced persistent symptoms for more than three months, with fatigue/exhaustion, concentration/memory problems and shortness of breath being most frequently reported. Risk factors were older age, female gender, previous illness, many and severe symptoms during the acute infection, and outpatient medical care. An impaired health-related quality of life was found in participants suffering from persistent symptoms. Overall, a high need for rehabilitation to improve health and work ability is evident. Further follow-up surveys will observe the changes and the impact of vaccination on the consequences of COVID-19 among health workers.

## 1. Introduction

The novel Severe Acute Respiratory Syndrome Coronavirus 2 (SARS-CoV-2) was initially discovered in China in late 2019, spread globally in a very short time, and has since been associated with significant morbidity and mortality. To date, over 500 million cases worldwide have been reported by the World Health Organization (WHO) [1]. The first case in Germany was identified in January 2020 [2], and since then, more than 23 million cases have been confirmed [3].

In an occupational context, health workers are more likely to be affected by COVID-19, an infection caused by SARS-CoV-2, compared with other occupational groups [4,5]. Nevertheless, there can be large differences within occupational groups and between different institutions [6]. Nursing staff in elderly care facilities and hospitals were particularly affected during the first wave in Germany [7]. By the end of 2021, a total of around 132,000 suspected cases of COVID-19 infection due to occupational causes have been reported to the Institution for Statutory Accident Insurance and Prevention in the Health and Welfare Services (BGW), and a little under 87,000 cases have been recognised as an occupational disease. This mainly concerned staff in inpatient and outpatient care as well as in hospitals [8].

What follows the acute phase of a SARS-CoV-2 infection has been gaining more attention. By now, there are numerous reports of long-lasting symptoms after a COVID-19 infection, in which those affected state mild to severe health impairments. All symptoms lasting longer than four weeks after the infection have been given the name “long COVID” or “post-acute sequelae”. If the symptoms persist for more than three months and cannot be explained by another condition, this is referred to as “post-COVID-19 syndrome (PCS)” [9,10].

The prevalence of long COVID varies between individual studies depending on the definition used, population, follow-up duration, and the symptoms and complaints studied. In a systematic review of working-age patients, incidences between 16 and 87% were reported [11]. Studies investigating the progress following a COVID-19 infection show that not only patients after a hospital stay but also non-hospitalised individuals can suffer from a variety of symptoms over a prolonged period of time and encounter restrictions in their daily activities. Persistent symptoms after a primarily mild course of infection were still present in 13% [12] or one-third of the study participants [13] seven months after diagnosis. The most frequently reported symptoms were taste and smell disturbances, shortness of breath and fatigue [12,14]. Other frequently reported symptoms included headache, concentration difficulties, exhaustion and reduced quality of life [15,16,17]. Long-term effects have also been observed in the occupational context for those affected. For example, self-reported long-COVID symptoms have been particularly common in educational, social and health professions in the UK [18].

Long-term studies of health workers suffering from the consequences of SARS-CoV-2 infection are limited. At the beginning of the study, little was known about the natural course of long-COVID and only a few studies focused on health and social workers. Using exposed working groups as reference cohorts instead of hospitalised patients allows for a more realistic estimation of the burden of long-COVID in these working groups. Being covered by a social insurance and compensation board for work-related accidents and diseases might influence the course of a disease or the related symptoms. However, this was not part of our consideration for the study. A difference to most other studies is that our cohort is younger, which influences the course of COVID-19.

Thus, a longitudinal study was initiated to investigate the consequences of infection for workers in health and social services over time. The aim of the study was to assess the frequency and duration of infection, the identification of risk factors for persistent symptoms and the impact of COVID-19 on health-related quality of life. This paper presents the results of the baseline survey.

## 2. Materials and Methods

### 2.1. Study Design and Study Population

The baseline survey is a cross-sectional study. Employees who are insured by the BGW with a suspected occupational COVID-19 infection were included as participants. The BGW is an accident insurance company for non-governmental health and welfare institutions in Germany. The requirement for the reporting was the suspicion of a job-related infection that was confirmed by polymerase chain reaction (PCR) testing and/or the presence of symptoms. In addition to health workers, the insured persons can also be social workers, and other employees in health or social facilities. Most employees are health professionals, and therefore, we use the term health workers in the following. The different occupations are described in Table 1.

Two regional administrations in eastern (Region 1) and western (Region 2) Germany were chosen. All insured persons from both regions with a COVID-19 infection reported before 31 December 2020 were included in the study.

In February 2021, a total of 4325 insured persons were contacted and informed about the study objectives, the study procedure and data protection. The exclusion criteria included the absence of SARS-CoV-2 infection, limited literacy skills and lack of German language skills. All participants gave their written consent to voluntarily participate in the study after a detailed briefing. A reminder letter was sent in April, asking the insured employees to participate. A short non-responder questionnaire was also attached with information on the course of symptoms and the reason for non-participation. The study was approved by the Ethics Committee of the Hamburg Medical Association (2021-10463-BO-ff).

### 2.2. Questionnaire

The participants were asked to fill out a comprehensive questionnaire on various topics for the purpose of data collection. Information was collected on socio-demographic data, physical activity, smoking status, height and weight, as well as information on occupation and scope of activity. Questions were also asked about the test for SARS-CoV-2 (PCR and/or antibody test) and the date of the test. The symptoms surveyed refer to the participant’s condition at the time of the acute infection and at the time of the interview. The information on acute and persistent symptoms was provided in a table to be ticked off by participants, who were asked to distinguish between not applicable, mild, moderate and severe degrees of severity. Fields that were not filled out were considered as not applicable. Further information on the course of the COVID-19 disease regarding the treatment of the acute infection (outpatient or inpatient) and any rehabilitation measures that were carried out were also collected. The Work Ability Index questionnaire [19] was used to assess both pre-existing illnesses and the capacity to work, as well as the subjective state of health. Work capacity and health status were reported by participants on a scale of 0–10 (0 = very poor, 10 = very good) for the period before the COVID-19 disease and at the time of the survey. The Multidimensional Fatigue Inventory scale (MFI) [20] was used to assess the general state of fatigue, and the Veterans RAND 12-Item Health Survey (VR-12) [21] was used to assess health-related quality of life. Information on depression and anxiety symptoms was collected using the Patient Health Questionnaire for Depression and Anxiety-4 (PHQ-4) [22].

### 2.3. Outcome

The main outcomes concern the long-term consequences of COVID-19 in health workers. These consequences are represented by persistent symptoms and symptom duration. In line with the National Institute for Health and Care Excellence (NICE) definition, all symptoms persisting beyond 4 weeks after the acute phase of SARS-CoV-2 infection are generally referred to as long COVID symptoms in this paper. If the symptoms last longer than 12 weeks or 3 months, they are referred to as PCS [9].

### 2.4. Statistical Analysis

The results for the metric variables are presented using the mean, median and standard deviation (SD). The categorical variables are expressed as absolute and relative frequencies. Group differences were analysed using Fisher’s exact test for categorical variables and a *t*-test for continuous variables. The missing data were taken into account and indicated in the tables. A binary logistic regression model was used to identify risk factors for persistent symptoms, and odds ratios (OR) with associated 95% confidence intervals (CI) were calculated. For this purpose, symptoms lasting longer than 3 months (PCS) versus no symptoms were defined as the dependent variable. The selection of variables depended on bivariate analysis. The inclusion criterion was a *p*-value of <0.1. The correlations between the individual predictors were tested for multicollinearity. An r of <0.8 was considered unproblematic. A *p*-value of ≤0.05 was considered statistically significant. SPSS (version 27, SPSS Inc., Armonk, NY, USA) was used to carry out the analyses.

## 3. Results

A total of 2053 insured persons (47.5% response) took part in the first survey, among whom 39% came from Region 1 and 61% came from Region 2 (Figure 1). A total of 554 (12.8%) insured persons were excluded due to wrong or missing address, absence of SARS-CoV-2 infection, poor literacy ability of refusal to participate. Based on the data from the non-responder analysis, the reasons given for non-participation included the absence of symptoms, feeling too sick, the long time since the infection or the fact that recovery has occurred in the meantime, and a lack of time or interest in the survey.

The median age of the study participants was 51 years, and 55% were 50 years and older. Nearly 82% of the participants were women, 84% were non-smokers, and 30.5% indicated they were not physically active (Table 1). Among non-responders, the median age was 42 years and 75% were women (no table). The median body mass index (BMI) was 25.8, and 56.9% of the participants were overweight or obese, according to the WHO obesity classification. A total of 16.3% lived alone in a household. More than half of the respondents (60.4%) worked as nurses, almost 10% worked as doctors and a minor proportion worked in other professions. Most of the participants were employed in a hospital (41.9%) or in residential care for older adults (35.4%), but other areas such as disability care, doctor’s practices or outpatient care were also frequently mentioned (Table 1).

Region 1 had a higher proportion of people over 50, and Region 2 had a higher proportion of people under 35. Full-time occupation was more common in Region 2, and part-time occupation more common in Region 1 (no table).

### 3.1. COVID-19 Infection

Half of all participants (51%) became infected with SARS-CoV-2 in the first half of 2020. In Region 2, the majority of participants became infected in the first half of the year, and in Region 1, almost 70% became infected in the second half of the year. A PCR test was performed on 97.1% of all participants, and 8.5% reported that an antibody test was performed (no table). A total of 94.7% of participants reported having symptoms during the acute phase of infection. The most common symptoms according to severity are shown in Figure 2. A total of 74.5% reported experiencing at least one severe symptom.

The most frequently mentioned acute symptoms were fatigue/exhaustion, headache and joint/limb pain, loss of sense of taste/smell, cough, concentration/memory problems and shortness of breath (Figure 2). Symptoms indicated as severe primarily included fatigue/exhaustion, loss of sense of smell/taste, joint/limb pain and headache. In total, 135 (6.6%) participants were treated for COVID-19 in the hospital, among whom 35 (1.7%/2053) received intensive care, and of these, 13 (0.6%/2053) required ventilation. Until the survey, 5.2% of the participants (n = 107) had not returned to work after the illness. Of these, 31% were hospitalised during the acute COVID-19 infection. Severe acute symptoms were reported by 81%, and severe persistent symptoms were reported by 60% (no table).

### 3.2. Long COVID/Post-COVID

At the time of the survey, 74.2% (n = 1523) of participants reported ongoing symptoms since their SARS-CoV-2 infection. With regard to the duration of symptoms, it was found that 50% of those affected have been suffering from the consequences of the disease for nine months or longer (max. 15 months). The most frequently reported long COVID symptoms were fatigue/exhaustion, concentration/memory problems and shortness of breath. Lack of motivation (32.0%), sleep disturbances (30.1%), hair loss (17.2%) dizziness (14.5%), cardiovascular problems, psychological stress such as anxiety and depression, and skin symptoms were also reported (no table).

The following evaluations refer to post-COVID sequelae of symptoms persisting for longer than three months. This was conducted by examining the PCS participants in comparison with those who did not have any symptoms at the time of the interview. Of the original 2053 participants, 123 were excluded due to missing data (n = 40) or a symptom duration shorter than three months (n = 83). A total of 1930 participants were included in these evaluations, of whom 1406 (72.8%) reported ongoing symptoms (Figure 1). One-third described at least one severe symptom. The frequency of persistent symptoms lasting longer than three months is shown in Figure 3 in comparison with the acute symptoms for the entire study population. The most common post-COVID symptoms were fatigue/exhaustion, concentration/memory problems, shortness of breath, headache and loss of sense of taste/smell. Other symptoms such as diarrhoea, nausea, fever, cold and sore throat were reported significantly less often for PCS compared with the acute phase. The persistent symptoms in the different degrees of severity of PCS are shown in the Appendix A.

The PCS group was older than the control group (median 52 vs. 47 years) and had a higher proportion of women (84.9% vs. 73.1%) (Table 2). Those affected were more often obese and had a pre-existing illness. Almost all PCS participants reported having not only a symptomatic infection but also multiple acute symptoms (median 8 vs. 5), which were often severe. PCS participants more frequently received inpatient treatment, and those affected were more often in need of intensive care and/or ventilation compared with the control group. Outpatient medical care was reported by almost 38% compared with 11%, and almost 48% desired rehabilitation compared with 10%. Smoking status, infection time and the region studied had no influence on persistent symptoms.

Outpatient care was shown to be a risk factor for symptoms lasting longer than three months, with an OR of 3.2 (95% CI 2.3–4.4) (Table 3). Other statistically significant risk factors were age from 50 years (OR 1.5; 95% CI 1.1–2.1), female sex (OR 1.6; 95% CI 1.2–2.2), pre-existing illness (OR 1.7; 95% CI 1.3–2.1), as well as the number and severity of symptoms during the acute infection phase (OR 1.2 and 1.6). The correlations between predictors were low (r < 0.70), indicating that multicollinearity did not affect the analysis.

The health-related quality of life for physical and mental health was rated lower by the post-COVID group compared with the asymptomatic group (Table 4). For the assessment of psychological distress, the total score of the PHQ-4 showed higher distress for one-fifth of the participants with PCS. Reports of depression or anxiety symptoms were considerably more often associated with persistent symptoms. The pre-infection assessment of personal work ability and health status had mean scores above 9 for both groups. Reduced scores were especially reported by participants with persistent symptoms regarding the current status (<7 vs. >8).

## 4. Discussion

This paper presents the results of the baseline survey of a longitudinal study of 2053 health workers who had a SARS-CoV-2 infection in 2020. Almost three-quarters of this group continued to suffer from the consequences months after the acute infection. Risk factors influencing symptoms persisting for longer than three months were older age, female gender, medically diagnosed pre-existing illnesses, a high number and severity of acute symptoms, and outpatient medical care. The health-related quality of life and the subjective work ability demonstrate significantly worse outcomes for people suffering from PCS compared with participants without symptoms at the time of the survey.

Many studies on long-term effects have now been conducted, especially for hospitalised patients [16]. However, only a few studies on health workers with a mild disease course have been conducted, and their objective, design, sample size and observation period vary widely. However, their results are similar to that of our study, where fatigue/exhaustion, concentration/memory problems, shortness of breath, headache and loss of taste/smell were most frequently reported as long-lasting symptoms. In an English study on hospital staff, fatigue, in particular, but also shortness of breath, anxiety and sleep disturbances were reported as long COVID symptoms [23]. Almost one-third of the staff of a Swiss hospital reported that they had not recovered their full health even after 90 days following a mostly mild SARS-CoV-2 infection. The most common complaints were fatigue, loss of smell/taste, general weakness and concentration problems [24]. Similarly, a Swedish study found the proportion of hospital staff who still had moderate to severe symptoms to be 26% after at least 2 months and 15% after 8 months, with ageusia, anosmia, fatigue and dyspnoea being the most commonly reported symptoms [25]. In a study of staff of various healthcare facilities in Denmark, the most common symptoms lasting longer than 3 months were dyspnoea, loss of taste/smell, muscle/joint pain and fatigue [26]. A high prevalence of symptoms lasting up to 90 days among non-hospitalised hospital staff was mainly found for taste and smell disturbances, and a slightly lower prevalence was found for dyspnoea compared with staff who were PCR-negative [27].

### 4.1. Risk Factors

Risk factors for long-lasting symptoms have mainly been described as female gender, age, high BMI, the severity of acute infection, number of acute symptoms, and various physical or mental pre-existing conditions [12,28,29,30,31,32,33]. Among health workers, older age; female gender; and in particular, pre-existing lung conditions, depression or level of exhaustion have also been identified as influencing factors [24,27]. In our study, age over 50 years, female gender, pre-existing illnesses and acute symptoms were also found to be correlated with longer persisting symptoms. However, a high BMI had no influence on persistent symptoms. The number of severe cases of acute COVID-19 due to hospitalisation was low in our study population, at just under 7%, and hence, no increased risk was found. In contrast, outpatient medical care during the acute phase of infection had a considerable influence, with an OR of 3.2 (95% CI 2.3–4.4). A further analysis of the outpatient cases showed that they did not have more acute symptoms, but the symptoms were often more severe. Those affected more often sought medical help and consulted other specialists besides general practitioners, especially pulmonologists, cardiologists, neurologists, ENT specialists or dermatologists. Occupation or scope of activity had no influence on the duration of symptoms after a COVID-19 infection.

### 4.2. Quality of Life

Health-related quality of life is a multifaceted concept that can be used to describe well-being and functional capacity. For example, a meta-analysis showed that after COVID-19, fatigue can significantly worsen the quality of life but is also associated with other symptoms such as dyspnoea, anosmia, sleep disturbances and impaired mental health [34]. For our study population, the assessments by Haller et al. [35] were also able to show that fatigue after SARS-CoV-2 infection is associated with a reduced quality of life, increased psychological distress, lower subjective health status and more frequent ability to work. Overall, our results indicate that health-related quality of life is considerably reduced for both physical and mental health in those with PCS. One-fifth of the respondents with persistent symptoms in our study reported a prevalence of depression and anxiety symptoms, which is comparable with a survey of 3678 predominantly non-infected doctors, nurses and medical technical assistants (MTA) [36]. A high level of psychological distress has been observed in everyday working life since the beginning of the pandemic, especially for those in the health professions, in addition to the physical stress. The experience of stress due to traumatic events and working on the front-line of COVID-19 [37] has been described in studies where it manifested itself primarily through symptoms of depression and anxiety [38,39]. The extent to which these occupational situations as well as the pandemic experience additionally affected the psychological state during COVID-19 infection or its consequences for the respondents cannot be assessed in our study, but this should be included in the consideration.

### 4.3. Strengths and Weaknesses

Our study was conducted with a large number of participants, comprising more than 2000 health workers from two different regions in Germany. Currently, there is only a limited number of studies available on the long-term consequences of SARS-CoV-2 infection for this group of people. By recruiting participants via an accident insurance provider, it was possible to study the consequences of the disease for very different occupational groups across different institutions. The high rate of response of just under 50% can be considered very good for occupational studies, which may be attributed to the novelty of the virus and to the high level of suffering in some cases. Nevertheless, a selection bias must be assumed. The non-responder analysis showed that the mean age of the non-participants was lower than that of the study population (42 vs. 51 years), whereas there was no difference in the gender ratio. Furthermore, it can be assumed that insured persons without persistent symptoms or with an asymptomatic infection may well have participated less frequently in the survey, and thus, an overestimation of the frequency of post-COVID in this cohort must be assumed. It is unclear why there is a prevalence of almost 75% with persistent symptoms in a middle-aged population that can generally be assessed as healthier due to their ability to work and who reported few severe acute COVID-19 diseases. Perhaps other diseases were partially responsible for the symptoms that have been falsely attributed to the COVID-19 infection.

In our study, we found a higher risk of post-COVID for women. This was also seen in other studies [24,27]. However, men are underrepresented in our study, so no generalised conclusion can be drawn. This needs to be confirmed in further studies with a larger group of men. Another limitation is data collection through the written survey. Self-reporting and self-assessment of symptoms are subjective and cannot be verified clinically. With regard to psychosocial aspects such as the health status and the perceived work ability, a bias cannot be ruled out when comparing the data with the retrospective assessment of the situation before the disease that was collected at the same time. The lack of an adequate control group without SARS-CoV-2 infection or from the general population is also an important limitation.

## 5. Conclusions

In our study, the number of health workers suffering from the long-term consequences of COVID-19 infection can be considered high, even though only a small proportion were still unfit for work at the time of the survey. In line with comparable studies, typical persistent symptoms and risk factors for post-COVID-19 syndrome were found. This study also demonstrates the urgent need for rehabilitation measures among those affected so that they can achieve an improved quality of life in terms of their health and work ability. The subsequent follow-up surveys of the study participants will show how the situation of those affected evolves and what influence the vaccinations will have on the long-term consequences of the SARS-CoV-2 infection.

## Figures and Tables

**Figure 1 ijerph-19-06983-f001:**
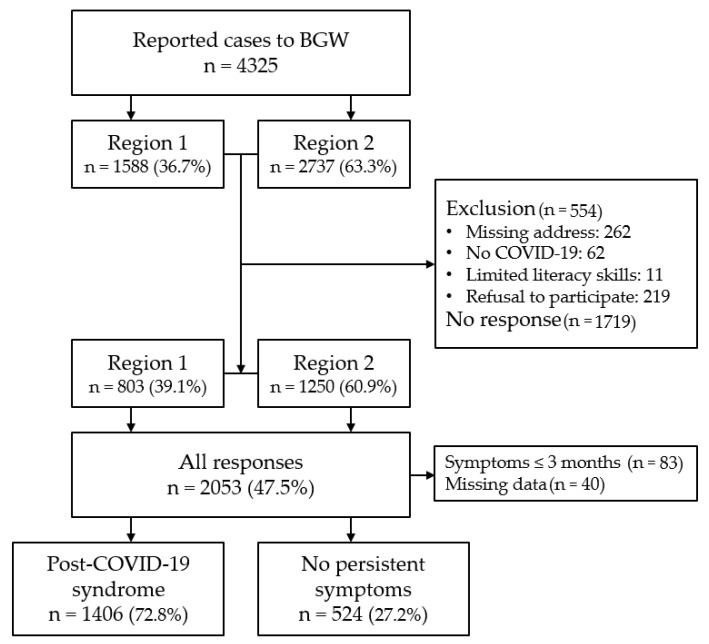
This flowchart describes the inclusion process of the study population and the categorisation into participants with persisting symptoms longer than 3 months (Post-COVID-19 syndrome) and without symptoms. BGW—Institution for Statutory Accident Insurance and Prevention in the Health and Welfare Services.

**Figure 2 ijerph-19-06983-f002:**
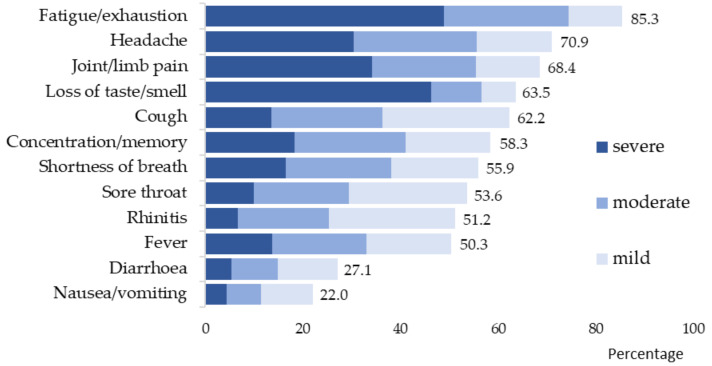
Acute symptoms of COVID-19 in the mild, moderate and severe degrees of severity (n = 1945/94.7%).

**Figure 3 ijerph-19-06983-f003:**
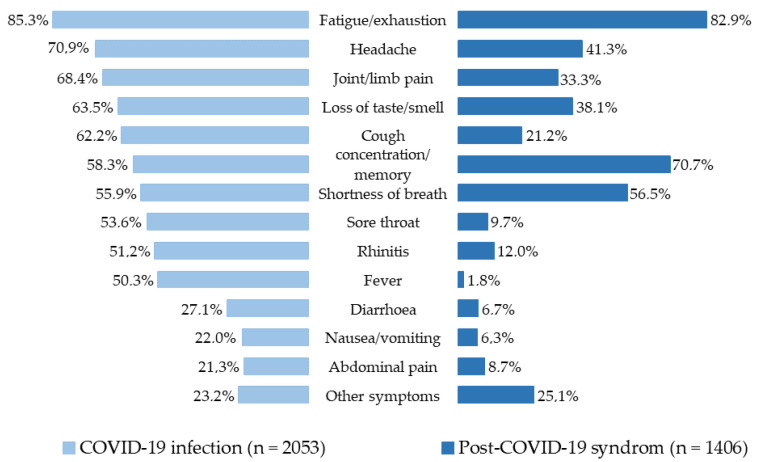
Acute symptoms and symptoms persisting longer than three months after COVID-19 (PCS) in health workers.

**Table 1 ijerph-19-06983-t001:** Characteristics of the study population (n = 2053).

Variables		n	%
Age, yrs.	<30	217	10.6
	30–39	327	15.9
	40–49	384	18.7
	50–59	744	36.2
	>59 years	381	18.6
Sex	Female	1677	81.7
	Male	376	18.3
Smoking	Smoker	327	16.1
(N/A = 22)	Non-smoker	1704	83.9
Physical activity	None	614	30.5
(N/A = 41)	1 h/week	486	24.2
	2–3 h/week	587	29.2
	>3 h/week	325	16.2
BMI	Underweight (<18.5)	31	1.5
(N/A = 31)	Normal weight (18.5–24.9)	841	41.6
	Pre-obesity (25.0–29.9)	667	33
	Obesity (≥30.0)	483	23.9
Family situation	Living alone	335	16.3
	Living with others	1718	83.7
Occupation	Nursing staff	1240	60.4
(N/A = 2)	Medical staff	201	9.8
	Therapeutic staff	121	5.9
	Housekeeping	112	5.5
	Social service	88	4.3
	Administrative staff	86	4.2
	Other	203	9.9
Workplace	Hospital	854	41.9
(N/A = 13)	Residential geriatric care	723	35.4
	Disability care	110	5.4
	Medical practice	95	4.7
	Outpatient care	82	4
	Other	176	8.6
Working time	Full-time	1004	49.1
(N/A = 10)	Part-time	947	46.4
	Other	92	4.5

N/A—not available/no answer.

**Table 2 ijerph-19-06983-t002:** Characteristics of participants with symptoms persisting more than 3 months (PCS) vs. no symptoms (n = 1930).

Variables		No Symptoms	PCS	*p*-Value
		(524/27.2%)	(1406/72.8%)	
		n/%	n/%	
Age	<35	132/25.2	229/16.3	<0.001
	35–49	157/30.0	362/25.7	
	>49 years	235/44.8	815/58.0	
Sex	Female	385/73.5	1194/84.9	<0.001
	Male	139/26.5	212/15.1	
Smoking	Smoker	85/16.4	218/15.6	0.7
Obesity	BMI ≥30	96/18.6	364/26.3	<0.001
Pre-existing disease		258/49.2	949/67.5	<0.001
Infection period during 2020	January–June	253/49.9	761/54.1	0.1
	July–December	254/50.1	645/45.9	
Region	Region 1	196/37.4	524/37.3	0.3
	Region 2	328/62.6	882/62.7	
Acute symptoms of COVID-19		447/85.3	1379/98.1	<0.001
No. of acute symptoms	Mean ± SD, Median	4.9 ± 3.4, 5.0	7.9 ± 3.2, 8.0	<0.001
Severe acute symptoms		273/52.1	1160/82.6	<0.001
Hospitalisation		14/2.7	119/8.5	<0.001
ICU treatment		2/0.4	33/2.3	0.002
Ventilation		0	13/0.9	N/A
Outpatient medical care		56/10.7	531/37.8	<0.001
Rehabilitation received		0	57/4.1	N/A
Rehabilitation request		46/9.8	616/48.9	<0.001

PCS—post-COVID-19 syndrome; N/A—not applicable.

**Table 3 ijerph-19-06983-t003:** Factors influencing symptoms persisting more than 3 months (PCS).

		No Symptoms	PCS	OR (95% CI)	*p*-Value	aOR (95% CI)	*p*-Value
		n/%	n/%				
Age	<35	132/25.2	229/16.3	Reference		Reference	
	35–49	157/30.0	362/25.7	1.1 (0.8–1.6)	0.5	1.2 (0.9–1.6)	0.3
	>49 years	235/44.8	815/58.0	1.5 (1.1–2.0)	0.01	1.5 (1.1–2.1)	0.004
Sex	Female	385/73.5	1194/84.9	1.7 (1.3–2.3)	<0.001	1.6 (1.2–2.2)	<0.001
	Male	139/26.5	212/15.1	Reference		Reference	
Obesity ^a^	BMI ≥ 30	96/18.6	364/26.3	1.1 (0.8–1.4)	0.7		
Pre-existing disease ^b^		258/49.2	949/67.5	1.6 (1.3–2.1)	<0.001	1.7 (1.3–2.1)	<0.001
Acute symptoms of COVID-19 ^b^		447/85.3	1379/98.1	1.5 (0.9–2.6)	0.2		
No. of acute symptoms	Mean ± SD	4.9 ± 3.4	7.9 ± 3.2	1.2 (1.1–1.3)	<0.001	1.2 (1.2–1.3)	<0.001
Severe acute symptoms ^b^		273/52.1	1160/82.6	1.6 (1.2–2.1)	0.002	1.6 (1.2–2.2)	0.001
Hospitalisation ^c^		14/2.7	119/8.5	1.2 (0.7–3.0)	0.6		
ICU treatment ^d^		2/0.4	33/2.3	2.4 (0.5–12.4)	0.3		
Outpatient medical care ^e^		56/10.7	531/37.8	3.2 (2.3–4.4)	<0.001	3.2 (2.3–4.4)	<0.001

OR—odds ratio; aOR—adjusted odds ratio; CI—confidence interval; PCS—post-COVID-19 syndrome. ^a^ no obesity as reference group; ^b^ not present as reference group; ^c^ no hospitalisation as reference group; ^d^ no ICU treatment as reference group; ^e^ no outpatient medical care as reference group.

**Table 4 ijerph-19-06983-t004:** Health-related quality of life in participants with symptoms persisting more than 3 months (PCS) vs. no symptoms (n = 1930).

Variables		No Symptoms	PCS	*p*-Value
		(524/27.2%)	(1406/72.8%)	
		n/%	n/%	
**Health-Related Quality of Life (VR-12)**				
Physical health	Range	20.0–61.5	11.7–65.0	<0.001
	mean (95% CI)	51.5 (50.9–52.0)	41.8 (41.3–42.3)	
Mental health	Range	5.4–67.3	7.3–66.5	<0.001
	mean (95% CI)	49.8 (50.1–51.6)	43.2 (42.5–43.8)	
**Psychological Stress (PHQ-4)**				
Total score	none/low	493/95.5	1118/80.9	<0.001
	moderate	15/2.9	195/14.1	
	strong	8/1.6	69/5.0	
Symptoms of depression	(≥3/6 points)	29/5.6	310/22.3	<0.001
Symptoms of anxiety	(≥3/6 points)	32/6.2	309/22.2	<0.001
**Subjective Work Ability**				
Before COVID-19	Mean ± SD	9.3 ± 1.3	9.3 ± 1.2	0.8
At the time of the survey	Mean ± SD	8.9 ± 1.7	6.8 ± 2.2	<0.001
**Subjective Health Condition**				
Before COVID-19	Mean ± SD	9.2 ± 1.1	9.1 ± 1.2	0.001
At the time of the survey	Mean ± SD	8.9 ± 1.4	6.9 ± 1.9	<0.001

PCS—post-COVID-19 syndrome; VR-12—Veterans Rand 12 Item Health Survey; PHQ-4—Patient Health Questionnaire 4.

## Data Availability

The data are available from the corresponding author upon request.

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
