# Peer review of "Long-Term Effects of COVID-19 on Workers in Health and Social Services in Germany"

_ijerph, 2022, doi:10.3390/ijerph19126983_

Round 1
Reviewer 1 Report
Thank you for the opportunity to review this article.
It addresses the consequences of coronavirus infection on the psychophysical functioning of health care workers. The authors surveyed a large group of workers with a self-report questionnaire.
I have no major objections to this work, but rather inquiries about the motivation for conducting the study.
When planning the study, did the authors assume that this group of subjects differed in any significant way from others previously studied in terms of the long-term consequences of coronavirus infection? Could the authors explain their research motivations better?
I also have a question about the regression analysis - were all variables entered in one model? If so, was this model statistically significant?
How do the authors explain the result that u sex (female) was a predictor of long-COVID? Some studies indicate that men die of COVID at a higher rate than women?
Can the authors cite the full survey in the paper (in the appendices)?
Reviewer 2 Report
Thank you for putting this manuscript together. The topic itself is of great interest with the increasing interest on long term covid and the impact of infection among different cohorts. I have identified various elements that require further review (from typos, figures and tables structure, wording) as well as a few aspects that are unclear from the reader's perspective. I have left these highlighted as well as comments included into the reviewed PDFs. Unfortunately I can only attach one file, reason why I have chosen the manuscript. However the table presented as supplementary material presents the same errors in title. Please amend.

Reviewer 3 Report
The article "Long-term effects of COVID-19 on health workers in Germany" is well written and the results and conclusion are supported by the data provided. However, there are few concerns
- the authors have mentioned the population under study as >2000, however, as per figure 1, 1403 subjects had long-term symptoms. There is a need to distinguish between long-term sequence vs with no persistent symptoms. I think, the authors are focusing on long-term symptoms, so please be consistent during description.
- The authors have mentioned that recruiting subjects from insurance company allowed people from different occupation but the title says health workers, please be specific.
Round 2
Reviewer 2 Report
The edits and additions to the manuscript are relevant and have improved the final version.